# Myeloperoxidase Alters Lung Cancer Cell Function to Benefit Their Survival

**DOI:** 10.3390/antiox12081587

**Published:** 2023-08-09

**Authors:** Nejra Cosic-Mujkanovic, Paulina Valadez-Cosmes, Kathrin Maitz, Anna Lueger, Zala N. Mihalic, Marah C. Runtsch, Melanie Kienzl, Michael J. Davies, Christine Y. Chuang, Akos Heinemann, Rudolf Schicho, Gunther Marsche, Julia Kargl

**Affiliations:** 1Division of Pharmacology, Otto Loewi Research Center, Medical University of Graz, 8010 Graz, Austria; 2BioTechMed-Graz, 8010 Graz, Austria; 3Department of Biomedical Sciences, Panum Institute, University of Copenhagen, DK-2200 Copenhagen, Denmark

**Keywords:** myeloperoxidase (MPO), A549 cells, non-small-cell lung cancer (NSCLC), neutrophil, proliferation, apoptosis

## Abstract

Myeloperoxidase (MPO) is a neutrophil-derived enzyme that has been recently associated with tumour development. However, the mechanisms by which this enzyme exerts its functions remain unclear. In this study, we investigated whether myeloperoxidase can alter the function of A549 human lung cancer cells. We observed that MPO promoted the proliferation of cancer cells and inhibited their apoptosis. Additionally, it increased the phosphorylation of AKT and ERK. MPO was rapidly bound to and internalized by A549 cells, retaining its enzymatic activity. Furthermore, MPO partially translocated into the nucleus and was detected in the chromatin-enriched fraction. Effects of MPO on cancer cell function could be reduced when MPO uptake was blocked with heparin or upon inhibition of the enzymatic activity with the MPO inhibitor 4-aminobenzoic acid hydrazide (4-ABAH). Lastly, we have shown that tumour-bearing mice treated with 4-ABAH had reduced tumour burden when compared to control mice. Our results highlight the role of MPO as a neutrophil-derived enzyme that can alter the function of lung cancer cells.

## 1. Introduction

Non-small-cell lung cancer (NSCLC) is a chronic inflammatory disease [1] that accounts for 85% of all lung cancer cases and has a high annual incidence rate [2,3]. Despite recent improvements in treatments for patients with NSCLC, the 5-year survival rate remains at only ~15% [4]. A better understanding of the complex tumour microenvironment (TME) would help to identify new strategies to improve treatment for patients with NSCLC.

Neutrophils are important innate immune cells that respond to inflammation [5] and represent a highly abundant immune cell type within the TME of NSCLC [6]. Pro- [7,8] and anti-tumorigenic [9] properties of neutrophils have been reported; however, tumour infiltrating neutrophils have been correlated with a poor clinical outcome [10]. Upon activation, neutrophils can release their content into the TME either by degranulation or by formation of neutrophil extracellular traps (NETs) [11]. Some proteins stored in the granules of neutrophils, such as neutrophil elastase (NE) and matrix metalloproteinase 9 (MMP-9), have been associated with tumour development [12,13]. Growing evidence has implicated myeloperoxidase (MPO), another granule protein, in the pathophysiology of different diseases including cancer [14]. 

MPO is a highly positively charged peroxidase enzyme that is mainly produced by neutrophils [15,16]. Some subpopulations of macrophages can also produce MPO, but at lower levels [17]. This enzyme utilises hydrogen peroxide (H_2_O_2_) as a substrate to oxidize halide and pseudohalide ions to produce hypohalous acids, such as hypochlorous acid (HOCl) [17,18]. Most effects of MPO are linked to its enzymatic activity and the generation of highly reactive products. These products can react and modify proteins, DNA, lipids and other oxidizable groups to cause cellular changes and genetic mutations [18,19,20]. However, there are also reports showing the effects of MPO on different cells, independent of its enzymatic activity [21]. Moreover, MPO has been shown to bind and internalise into different cells including platelets [22], endothelial cells [23,24], macrophages [25] and epithelial cells [26], thereby altering signalling pathways [15,21].

In recent years, the role of MPO in cancer has received attention, despite being a relatively new area of research. Several studies have noted the involvement of MPO in the regulation of cancer. While MPO has both pro- and anti-tumour properties, most of the evidence suggests that it supports tumour initiation and progression [14]. MPO has been found to contribute to tumour initiation by creating a hypermutagenic environment through the oxidation and modification of DNA by MPO-derived oxidants [26,27]. In addition, MPO is also implicated in cancer progression, as it affects tumour growth, apoptosis, cell migration, and metastasis [14]. The effects of MPO in lung cancer development are still unclear. However, studies have associated low MPO expression, due to a polymorphism in the human MPO gene, with a reduced risk of lung cancer [28]. 

The purpose of the current study was to determine the role of MPO in lung cancer cells. Using the lung adenocarcinoma cell line A549, we showed that MPO promotes cancer cell proliferation while protecting cells from apoptosis. Moreover, MPO internalized into A549 cells and was found in the nuclear fraction. Some of the functions of MPO in cancer cells were abolished by using an MPO inhibitor or by blocking the internalization of the enzyme into the cells. Our results point to MPO as a neutrophil-derived enzyme that positively impacts the behaviour of lung cancer cells.

## 2. Materials and Methods

### 2.1. Cell line and Cell Culture 

All experiments were performed using the human lung adenocarcinoma cell line A549 obtained from ATCC (ATCC^®^ CCL-185™). Cells were maintained in DMEM media supplemented with 10% Fetal Bovine Serum (FBS, all Life Technologies, Vienna, Austria) and 1% penicillin/streptomycin (P/S, PAA Laboratories, Pasching, Austria) at 37 °C, 5% CO_2_ in a humidified atmosphere. All treatments were performed in DMEM without FBS or PS. For in vivo experiments, the murine KP cell line was generously provided by Dr. McGarry Houghton (FredHutch, Seattle, WA, USA) [29]. KP cells were maintained in DMEM with 10% FBS and 1% P/S at 37 °C, 5% CO_2_ in a humidified atmosphere. 

### 2.2. Cell Treatment

Lyophilized MPO (Elastin Products Company, Owensville, MO, USA) was diluted in ddH_2_O to a concentration of 1 mg/mL and stored at 4 °C. MPO was used in the following concentrations: 0.5 µg/mL (3.33 nM), 2 µg/mL (13.33 nM), 5 µg/mL (33.33 nM), 10 µg/mL (66.67 nM) or 20 µg/mL (133.33 nM). The MPO inhibitor 4-ABAH (Sigma-Aldrich, St., Saint Louis, MO, USA, #475994) was dissolved in 100% of DMSO to a concentration of 20 or 40 mM. Heparin 1000 I.E./mL from Gilvasan (#3909969) was used directly from the bottle. R19-S (Futurechem, Seoul, Republic of Korea; #FC-8001) was prepared as a 1 mM stock in acetonitrile and stored at 4 °C.

### 2.3. Bromodeoxyuridine (BrdU) Cell Proliferation Assay

A total of 2.5 × 10^5^ A549 cells were seeded in 6-well plates and incubated for up to 24 h at 37 °C. Cells were serum starved for 4 h, followed by treatment with different concentrations of MPO (0.5, 2, 5, 10 µg/mL) for 24, 48 and 72 h. In addition, ddH_2_O was used as vehicle control. For inhibitor experiments, cells were incubated for 30 min with 10 µM 4-ABAH prior to MPO treatment (5 µg/mL) and incubated for 48 h. A total of 10 µL of 1 mM BrdU-Solution (BD Bioscience, Franklin Lakes, NY, USA; #559619) was added to cells in media 4 h before detaching to allow BrdU to be incorporated into the newly synthesized DNA. Cells were detached from the plate with 300 µL accutase (PanBiotech, Aidenbach, Germany) and collected in 5 mL FACS tubes. The staining was performed without 7-AAD according to the instructions provided by the company. In brief, cells were fixed for 15 min with BD Cytofix/Cytoperm followed by 10 min incubation with BD Cytoperm Permeabilization Buffer Plus. After 5 min incubation with BD Cytofix/Cytoperm, cells were incubated with DNase for 1 h at 37 °C to expose the incorporated BrdU. Anti-BrdU antibody was prepared in a 50 µL 1× Perm Wash and incubated with cells for 20 min at room temperature. Samples were measured using a FACS Canto II flow cytometer (BD Biosciences).

### 2.4. Ki67 Cell Proliferation Assay 

A total of 2.5 × 10^5^ A549 cells were seeded in 6-well plates and incubated for up to 24 h at 37 °C. Cells were serum starved for 4 h, followed by treatment with 10 µg/mL MPO for 72 h. In addition, ddH_2_O was used as the vehicle control. Cells were detached with accutase and collected in falcon tubes. After washing cells with Dulbecco’s Phosphate Buffered Saline (PBS, Gibco, Fisher Scientific, Loughborough, UK), fixation and permeabilization were achieved by adding 3 mL ethanol (−20 °C) to the cell pellet, drop by drop, while vortexing. After incubating for 1 h at −20 °C, 9 mL of staining buffer (SB; PBS + 2% FBS) was added to each sample and centrifuged at 500× *g* for 5 min at 4 °C. Cells were stained with 3 μL/test of Ki67-PerCP/Cy5.5 (Biolegend, San Diego, CA, USA) in 100 μL SB and incubated for 30 min at room temperature. Afterwards, 200 μL of SB was added to each sample and centrifuged, and cell pellets were resuspended in 200 μL SB. Samples were analysed within 1 h using a FACS Canto II flow cytometer. 

### 2.5. MTS Cell Proliferation Assay

A total of 5 × 10^3^ A549 cells were seeded in a 96-well assay plate and incubated at 37 °C for up to 24 h. Afterwards, cells were washed 1× with PBS and starved for 4 h in DMEM without FBS and PS. Cells were then treated by triplicates with different concentrations of MPO (0.5, 2, 5, 10 μg/mL) and incubated for 48 h. In addition, ddH_2_O was used as the vehicle control. A total of 10 μL of thawed MTS Solution (Promega, Walldorf, Germany) was added directly to cells and incubated for 60 min at 37 °C. The produced amount of soluble formazan was measured at 490 nm using a xMark Microplate Spectrophotometer (Biorad, Vienna, Austria).

### 2.6. Annexin V and Propidium Iodide (PI) Apoptosis Assay

A total of 5 × 10^5^ A549 cells were seeded in 6-well plates, grown for up to 24 h and serum starved overnight. Cells were then treated with different concentrations of MPO (0.5, 2 and 5 μg/mL) for different durations (3, 6, 15 and 24 h). Otherwise, cells were incubated for 1 h with heparin (150 µg/mL) and treated with 5 µg/mL MPO for 15 h. Afterwards, cells were detached using accutase, collected in 5 mL FACS tubes and washed with PBS. The cell pellets were resuspended in 100 μL apoptosis-staining solution [2 μL Annexin V and 2 μL propidium iodide (PI) in 100 μL 1× Binding Buffer/test] (BD Bioscience, Franklin Lakes, NY, USA) and incubated for 15 min in the dark at room temperature. A total of 100 μL of 1× Binding Buffer was added to each sample and analysed within 1 h using a FACS Canto II flow cytometer. The percentages of non-apoptotic cells (AnV−/PI−), early apoptotic cells (AnV+/PI−), late apoptotic cells (AnV+/PI+) and necrotic cells (AnV−/PI+) were recorded.

### 2.7. Western Blot Analysis 

A total of 1–2 × 10^6^ cells were seeded in a 10 cm dish and incubated for up to 24 h. Cells were then serum starved for 2–4 h or overnight, followed by MPO treatment for the indicated durations. Heparin was incubated for 1 h prior to MPO treatment. Cells were washed with PBS and snap frozen in liquid nitrogen (stored at −80 °C until use) or used fresh for protein lysates.

Total protein lysates: Cells were scraped off of the plates and lysed using IP buffer (0.1% Triton X-100, 150 mM NaCl, 25 mM KCl, 10 mM Tris HCl, pH 7.4; 1 mM CaCl_2_ in H_2_O) containing a 1:100 protease/phosphatase inhibitor cocktail (Cell Signaling, Danvers, MA, USA). After centrifugation at 18,620× *g* for 10 min at 4 °C), the total protein concentration was determined using a Pierce BCA assay (Thermo Scientific, Waltham, MA, USA). Furthermore, 4× NuPage sample buffer (90 µL 4× LDS sample buffer + 10 µL reducing reagent) was mixed with the samples and boiled for 10 min at 95 °C. 

Nuclear and cytoplasmic fractions: Detached cells were centrifuged at 16,060× *g* for 5 min at 4 °C and the cell pellet was resuspended in 500 µL Buffer A [10 mM HEPES, 10 mM KCl, 0.1 mM EDTA, protease/phosphatase inhibitor (100×), Cell Signaling] in PBS. Cells on ice were homogenized at medium speed for 15 sec, followed by the addition of 25 µL NP-40, and mixed by vortexing. After centrifugation at 2370× *g* for 5 min at 4 °C, the cytoplasmic fraction in the supernatant was transferred in a separate epi and kept on ice. To generate the nuclear cell fraction, the remaining cell pellet was mixed with 50 µL ice-cold buffer C (20 mM HEPES pH 7.9, 25% glycerol, 0.4 M NaCl, 1 mM EDTA in PBS without Ca^2+^ and Mg^2+^). The gently dislodged pellet was actively rocked for 30 min in ice and centrifuged at 18,620× *g* for 10 min at 4 °C. The nuclear fraction in the supernatant was collected in a separate Eppendorf tube. The same volume of both fractions was applied and analysed by Western blot analysis. 

Subcellular fractions: were generated using the Subcellular Protein Fraction Kit for Cultured Cells (Thermo Scientific, #78840) following the instruction manual. In brief, cells were incubated with the CEB reagent for 10 min on ice and centrifuged at 500× *g* for 5 min. The supernatant with the cytoplasmic fraction was collected in a pre-chilled tube. To dissolve the plasma, mitochondria and ER/golgi membranes, the MEB reagent was incubated with the remaining pellet for 10 min on ice. The nucleus was recovered by centrifugation at 500× *g* for 10 min and the membrane fraction in the supernatant was collected in a pre-chilled tube. The pellet was further incubated with NEB reagent for 30 min on ice to generate the soluble nuclear extract. After centrifugation at 5000× *g* for 5 min, the nuclear extract in the supernatant was collected. The remaining pellet was incubated with NEB reagent and micrococcal nuclease (15 min, room temperature) to release chromatin-bound nuclear proteins. After centrifugation at 16,000× *g* for 5 min, the supernatant containing the chromatin-binding extract was collected in a separate tube. Samples were snap-frozen and stored at −80 °C until use.

Western blot analysis: 20 µg protein from homogenates was electrophoresed for 45 min at 200 V using the NuPage, Bis-Tris gel (Invitrogen, Waltham, MA, USA). To investigate MPO uptake with heparin, equal volumes of homogenates were loaded on the gel. Using the iBlot transfer device, proteins were transferred to a membrane and blocked for 1 h with TBST (1× TBS + 0.1% Tween) containing 5% milk. Membranes were incubated at 4 °C overnight with the primary antibody. The following monoclonal antibodies were used: Anti-MPO (rabbit anti-human MPO, clone: E1E7I, 1:500), anti-GAPDH (rabbit anti-human GAPDH, clone: 14C10, 1:5000), anti-Lamin A/C (mouse anti-human Lamin A/C, clone: 4C11, 1:500), anti-tAKT (rabbit anti-human tAktin, clone: 11E7, 1:1000), anti-pAKT (rabbit anti-human phospho-AKT (Ser473), clone: D9E, 1:1000), anti-AKT (rabbit anti-human AKT (pan), clone: 11E7, 1:1000), anti-ERK1/2 (p44/42 MAPK; rabbit anti-human ERK1/2, clone: 137F5, 1:1000), anti-pERK1/2 (phosphor-p44/42 MAPK (Thr202/tyr204); rabbit anti-human pERK1/2, clone: D13.14.4E, 1:1000), anti-Histone3 (rabbit anti-human histone3, clone: D1H2, 1:2000), anti-HSP90 (rabbit anti-human HSP90, clone: C45G5, 1:1000) and anti-EGFR (rabbit anti-human EGF Receptor, clone: D38B1, 1:1000). All antibodies were purchased from Cell Signaling, USA. Afterwards, the membranes were washed for 30 min and incubated for 2 h with the secondary antibody HRP-goat anti-rabbit IgG (polyclonal, 1:5000, Jackson ImmunoResearch, Philadelphia, PA, USA). Washed membranes were developed using ECL solution (BioRad, Hercules, CA, USA).

### 2.8. Immunofluorescence Staining of MPO

A total of 1 × 10^4^ or 2 × 10^5^ A549 cells were seeded in 4-well chamber slides or on coverslips in 6-well plates and incubated for up to 24 h at 37 °C. Cells were serum starved for 2–4 h followed by MPO incubation for the indicated timepoints. To investigate MPO uptake by A549 cells, the cells were incubated with heparin (150 µg/mL) for 1 h before adding MPO (10 µg/mL) for 2 h. Cells treated with ddH_2_O or DMSO (0.2%) were used as the controls. Cells were rinsed with PBS (3 × 2 min) before fixing and permeabilizing with ice-cold methanol for 20 min at −20 °C. Cells were washed with PBS (3 × 2 min on shaker) and incubated with blocking solution (5% goat serum + 5% BSA in PBS with 0.1% TritonX) for 60 min at room temperature (on shaker). The primary antibody (rabbit anti-MPO 1:500; Cell Signalling, Danvers, MA, USA) was prepared in an antibody diluent (1:10 blocking solution in PBS-TritonX) and incubated overnight at 4 °C. After washing the cells with PBS (3 × 2 min on shaker), they were incubated with the secondary antibody (goat anti-rabbit AF488, 1:500) for 1 h at room temperature in the dark. After washing the cells with PBS (5 × 2 min on shaker), they were mounted using VECTASHIELD with DAPI. The cells were visualized using an Olympus iX70 fluorescence microscope and the Olympus cellSens Dimension 2.3 imaging software with Z-stacking to allow imaging of different cell layers. 

### 2.9. Intracellular MPO Flow Cytometry Staining 

A total of 3.5 × 10^5^ A549 cells were seeded in a 6-well plate and incubated for up to 24 h. After serum starvation for 4 h, the cells were treated with indicated concentrations of MPO for 2 h. Otherwise, cells were treated with heparin (150 µg/mL) for 1 h prior to MPO treatment. Cells were detached using accutase and collected in FACS tubes for intracellular staining with a transcription factor buffer set (BD Bioscience, #56257). To exclude dead cells, FVD eFluor™ 780 (FVD, eBioscience) was incubated for 30 min at 4 °C. After 2× washing with SB, cells were fixed and permeabilised with 1X TF Fix/Perm Buffer solution for 40 min at 4 °C in the dark. Cells were washed twice with 1X Perm/Wash followed by incubation with 1.5 µL humanTruStain FcX blocking solution (FcBlock, Biolegend) in 1× Perm/Wash and incubated at 4 °C for 10 min. Subsequently, cells were stained with anti-MPO FITC antibody (Biolegend, #333138) for 20 min at 4 °C in the dark. After 2× washing with 1× Perm/wash and SB, the cells were analysed with a FACS Canto II flow cytometer. 

### 2.10. Basal Production of Hydrogen Peroxide (H_2_O_2_) by A549 Cells

The Intracellular Hydrogen Peroxidase assay kit (Sigma-Aldrich, Taufkirchen, Germany; MAK164) was used for imaging of H_2_O_2_ production. A549 cells (1.5 × 10^4^) were seeded into a black/clear-bottom 96-well plate and incubated for up to 24 h, followed by 4 h serum starvation. Cells were then incubated with 100 µL of the Fluorescent Peroxide Sensor (FITC) for 15 min and washed with PBS. To stain the nuclei, 1 drop of NucBlue™ Live ReadyProbes™ Reagent (Thermo Fisher, Hoechst 33342) was added to cells in PBS and they were visualized using an Olympus iX70 fluorescence microscope and the Olympus cellSens Dimension 2.3 imaging software. 

### 2.11. Measurement of HOCl Production Using the “R19-S” Sensor

A total of 2–2.5 × 10^5^ cells were seeded directly or on coverslips in a 6-well plate and incubated for up to 24 h. Afterwards, cells were serum starved for 2 h followed by MPO (5 µg/mL) treatment for 2 h or 3 h for microscopy. For the inhibitor experiments, the cells were first treated with 10 µM 4-ABAH for 30 min before adding MPO (5 µg/mL) for 2 h. Cells treated with ddH_2_O or DMSO (0.1%) served as the vehicle control. Cells were washed with PBS and incubated with 10 µM R19-S (Futurechem, Seoul, Republic of Korea; FC-8001) in serum-free media for 1 h. For signal quantification, cells were washed 2× with PBS, scraped off of the plate and transferred into a 96-well plate, and fluorescence was measured using a CLARIOstar microplate reader (λ_ex_ 514 nm; λ_em_ 530–603 nm). For flow cytometry analysis, cells were collected in FACS tubes, fixed using IC fixation buffer (Thermo Fisher) for 10 min at 4 °C, and the fluorescent signal was quantified using a FACS Canto II flow cytometer (PE channel). For imaging, cells on coverslips were washed 2× with PBS, and fixated and permeabilized with 3.7% paraformaldehyde (PFA) for 10 min at room temperature, followed by 6 min incubation with ice-cold methanol at −20 °C. Cells were rinsed with PBS and incubated in PBS up until the next day. Cells were mounted using VECTASHIELD with DAPI and visualized using an Olympus iX70 fluorescence microscope and the Olympus cellSens Dimension 2.3 imaging software with Z-stacking. 

### 2.12. Murine Tumour Model and ABAH Application

All animal experiments were performed at the animal facilities of the Medical University of Graz. The experimental protocols were approved by the Austrian Federal Ministry of Science and Research (GZ 2022-0.748.851). Wild-type C57BL/6J (B6) mice were purchased from Charles River, Germany. A total of 5 × 10^5^ KP cells suspended in 450 µL PBS (Gibco) were subcutaneously (s.c.) injected into the flank of 6–14-week-old MPO B6 mice. The injections were performed with mice under inhaled isoflurane anaesthesia. A total of 40 mg/kg 4-ABAH was injected intraperitoneally (i.p.) every day until the tumours were harvested. The control group (vehicle) was injected with 200 µL PBS containing 2% DMSO. Tumour growth was monitored and measured (length and width) every day using a digital calliper during the duration of the experiment. After 15 days, tumours were harvested, weighed, and measured with a digital calliper ex vivo. Tumour volume was calculated using the formula: v = length × width × height × π/6 [30].

### 2.13. Statistical Analyses

Statistical analyses were performed using GraphPad Prism 6.1 (GraphPad Software, La Jolla, CA, USA). Significant differences between the two experimental groups were assessed using unpaired Student’s *t*-tests with Welch´s correction; otherwise, a paired t-test was performed. For comparing three or more groups, one-way ANOVA was used with the Dunnett´s or Tukey´s post hoc test for correction of multiple comparisons. Results are presented as the mean ± SD where the *p*-value < 0.05 was considered as statistically significant and is shown as 1, 2, 3 or 4 asterisks when lower than 0.05, 0.01, 0.001 or 0.0001. 

## 3. Results

### 3.1. MPO Alters A549 Proliferation and Apoptosis In Vitro

Previous studies in our laboratory have shown that MPO-deficient mice (MPO KO; C57BL/6 MPO^−/−^) have up to a 40% reduction in tumour size compared to MPO wild-type mice (C57BL/6 MPO^+/+^) [29]. To further understand the role of MPO in the setting of lung cancer, we aimed to investigate whether MPO can alter the function of tumour cells in vitro. We exposed human lung adenocarcinoma cells (A549 cells) to different concentrations of MPO and assessed the effects of the enzyme on cell proliferation. A549 cells treated with MPO showed increased proliferation when compared to vehicle-treated cells (Figure 1A–C and Appendix A). Using the BrdU assay, an increase in proliferation of approximately 50% was observed after 48 h of MPO treatment (Figure 1A). A similar effect was also observed after 24 h and 72 h of MPO treatment (Appendix A). Elevated metabolic activity of MPO-treated A549 cells as an indirect measure of cell proliferation was confirmed by using the MTS assay (Figure 1B). In addition, we stained the cells with the proliferation marker Ki67, which is only expressed throughout the active cell cycle (G1, S, G2 and M phases) [31]. With this method we verified that MPO enhances cell proliferation after 72 h (Figure 1C). 

A role for MPO in the apoptosis of endothelial and epithelial cells has been reported [14]. For instance, MPO reduced apoptosis in epithelial ovarian cancer (EOC) cells by increasing S-nytrosylation of caspase-3 [32]. To test whether MPO impacts the apoptosis of A549 cells, we exposed them to increasing concentrations of MPO and performed Annexin V (AnV)/PI staining (Figure 2A–E and Appendix A). The strongest effect on apoptosis was seen after 15 h of MPO treatment, where MPO significantly elevated the percentage of live (AnV−/PI−) cells and reduced the percentage of dead (AnV−/PI+), early (AnV+/PI−) and late (AnV+/PI+) apoptotic cells (Figure 2A–E).

Our data suggest that MPO can alter the proliferation and apoptosis of A549 cells to benefit their survival. We therefore investigated the effect of MPO in the activation of intracellular signalling pathways that control cell growth, including the AKT and ERK pathways [33,34]. A549 cells were treated with MPO, and Western blot analysis was used to determine differences in the phosphorylation status of the AKT and ERK proteins. After 15 min of treatment, AKT and ERK phosphorylation was significantly higher in MPO-stimulated A549 cells as compared to unstimulated controls (Figure 3A,B). No changes were observed in the total amount of both proteins (Figure 3A,B). 

### 3.2. MPO Binds to and Is Taken Up by A549 Cells

It has been reported that MPO binds to and is internalised into epithelial cells [35]. We therefore investigated MPO uptake in A549 cells by treating them with different concentrations of MPO. Cells were washed before lysis and Western blotting was performed. MPO bound to the surface of A549 cells after 5 min of exposure, indicating a rapid uptake by cancer cells (Figure 4A). The time-dependent interaction of MPO with cells was confirmed with immunofluorescence staining of MPO-treated cells (Appendix A). Additionally, treatment with increasing concentrations of MPO enhanced the amount of MPO associated with cells in vitro (Figure 4B). Intracellular flow cytometry staining showed that the majority of A549 cells were positive for MPO after 2 h of exposure (Figure 4C and Appendix A). 

To further investigate the localisation of MPO after cellular uptake, immunofluorescence staining was performed. Besides the already reported cytoplasmic uptake, a minor MPO signal was also detected in the nuclear region after 2 and 24 h of MPO exposure, while untreated control cells showed no MPO signal (Figure 5A). To validate the nuclear translocation of the enzyme, A549 cells were treated with MPO for 2 h and cytoplasmic and nuclear extracts were obtained. To verify the purity of the fractions, Laminin A/C was used as a nuclear marker, whereas AKT served as a cytoplasmic marker. Using Western blot analysis, we identified bands assigned to the heavy chain of MPO (~60 kDa) in both the cytoplasmic and nuclear fractions of A549 cells (Figure 5B). As expected, MPO is not expressed endogenously by A549 cells, since we did not detect the MPO signal in the untreated samples (Figure 5B). Additional separation into subcellular fractions showed MPO was bound to the membrane and chromatin fraction of MPO-treated A549 cells (Figure 5C). Besides the rapid binding and uptake of MPO, we found that the enzyme persisted in cells (cytoplasmic and nuclear fraction) for up to 72 h after exposure to MPO (Appendix A). 

### 3.3. Blocking MPO Internalisation with Heparin Abolishes MPO Effects on Apoptosis in A549 Cells

The mechanism by which MPO binds and internalises into cells is still unclear. It has been reported that highly cationic MPO binds to the negatively charged parts of glycosaminoglycans (GAGs) on the surface of cells and internalises into cells [23,24]. A study on endothelial cells reported the co-localisation of MPO and heparan sulfate after MPO treatment, supporting the idea of charge-dependent binding and uptake [36]. Baldus et al. showed that treating cells with heparin could reduce MPO uptake by approximately 70% in endothelial cells [24]. In this regard, we investigated the mechanism of MPO uptake by A549 cells using heparin to block the binding and internalisation of MPO. Cells were treated with heparin for 1 h before exposing them to MPO and separating them into cytoplasmic and nuclear fractions for Western blot analysis. Heparin treatment inhibited MPO binding and uptake in A549 cells (Figure 6A). Using intracellular flow cytometry staining, we observed a 95% reduction in intracellular MPO after heparin treatment (Figure 6B). In addition, we visualised the reduction in MPO uptake with heparin via immunofluorescence staining for MPO (Figure 6C). 

To determine if MPO can alter the function of A549 cells without binding or entering the cells, we exposed cells to heparin prior to MPO treatment, with effects on apoptosis determined after 15 h of MPO treatment. Blocking MPO uptake with heparin abolished the protective role of MPO in apoptosis. We found no difference in the percentage of early apoptotic cells in untreated control vs. heparin + MPO-treated cells (Figure 6D). These data suggest the importance of the cellular association or uptake of MPO to induce functional changes in A549 cells. 

### 3.4. MPO Is Active after Internalising A549 Cells

MPO utilizes H_2_O_2_ in an oxidative reaction with halides to produce reactive oxygen species (ROS), such as HOCl [37]. To assess the enzymatic activity of MPO after cellular uptake, we first determined the availability of MPO substrate H_2_O_2_ in A549 cells. We found that H_2_O_2_ is produced by A549 cells under basal conditions, although the signal was not found in every cell. (Appendix A). Using the rhodamine 19 sensor (R19S), we investigated intracellular MPO activity. HOCl generated by MPO can react with R19S to form a highly fluorescent rhodamine 19 (R19), which can be used to measure MPO activity [38]. We treated A549 cells with MPO for 2 h, followed by 1 h incubation with R19S. Production of HOCl was detected using flow cytometry (Figure 7A) and via plate-based quantification of fluorescence (Figure 7B). With both methods, we observed higher signals following MPO treatment, suggesting that the enzymatic activity of MPO persists after internalisation into cells. Using fluorescence microscopy, the MPO signal was detected in the cytoplasm and was highly concentrated around the nuclei of A549 cells (Figure 7C). 

Next, we investigated whether the previously observed increase in proliferation of MPO-treated cells depends on the enzymatic activity of MPO. Using 4-ABAH, an irreversible inhibitor of MPO [39], we assessed the effect of MPO inhibition on A549 cell proliferation. The 4-ABAH pretreatment significantly reduced R19S fluorescence, indicating reduced enzymatic activity and HOCl production in A549 cells. (Figure 7D). After exposing cells to 4-ABAH and MPO for 48 h, there was a significant reduction in cell proliferation when compared to MPO treatment alone (Figure 7E and Appendix A). These data suggest the involvement of MPO activity in altering A549 cell function. 

### 3.5. 4-ABAH Reduces Tumour Growth In Vivo

We showed that MPO activity is involved in altering lung cancer cell function in vitro. Therefore, we investigated the effects of MPO inhibition on tumour growth in vivo. We thus treated KP tumour-bearing mice with 5.3 mg/mL 4-ABAH i.p. every day until sacrifice (Figure 8A). We observed that treatment with 4-ABAH led to slower tumour growth and reduced the tumour size when compared to the control group (DMSO 2%) (Figure 8B,C). 

Altogether, we found that enzymatically active MPO plays a role in promoting tumour development of lung cancer cells in vitro and in vivo. Nevertheless, further investigations on the involvement of the enzyme activity are necessary to confirm these findings.

## 4. Discussion

Neutrophils and neutrophil-derived molecules impact tumour development, especially favouring tumour growth. In this context, growing evidence shows the involvement of MPO in the pathophysiology of different diseases, including cancer [14]. However, little is known about the role of MPO in NSCLC. Low MPO expression has been linked to a reduced risk of developing lung cancer [28]. Moreover, different studies have indicated a pro-tumorigenic function of MPO in the lung TME [29,40]. Recent studies from our laboratory showed that tumour-bearing MPO knock-out mice (MPO KO) had reduced tumour growth when compared to their wild-type (WT) littermates [29]. Interestingly, MPO directly influenced CD8^+^ T-cell function in vitro and in vivo, thereby affecting tumour development [29]. Accordingly, we aimed to determine whether MPO can also influence cancer cell function and its mechanism of action. 

In this study, we used the human lung cancer cell line (A549 cells) with an epithelial origin to examine the role of MPO in cancer development. We found that in vitro MPO-treated A549 cells had increased proliferation and reduced apoptosis when compared to untreated control cells. Then, we investigated the activation of AKT and ERK signalling pathways as the major regulators of cell proliferation and survival. Indeed, after 15 min of MPO exposure, we observed an increased activation of these signalling pathways. These data could explain the mechanism by which MPO alters the function of A549 cells. It has been shown that MPO can interact with CD11b/CD18 (Mac-1 integrin) on the surface of neutrophils to suppress apoptosis by increasing the phosphorylation of AKT and ERK [41]. The impact of MPO on the AKT and ERK signalling pathways was also observed in endothelial cells [23]. Saed et al. showed that MPO reduced apoptosis in epithelial ovarian cancer cells (EOC) by nitric-oxide-induced S-nitrosylation of caspase-3 [32]. Furthermore, MPO was shown to reduce proliferation and inhibit wound healing in epithelial cells [42]. Therefore, we investigated the binding and cellular uptake of MPO by A549 cells. 

We found that MPO rapidly binds to the surface of lung cancer cells and that the amount of bound enzyme depends on the concentration/availability in the media. The intracellular uptake was observed in the majority (>60%) of cells. Interestingly, we found that MPO partially translocated into the nuclei of A549 cells. It was previously reported that MPO can be found in the nuclei of neutrophils during NETosis [43]. 

MPO binds to the cell surface via glycosaminoglycans (GAGs), dependent upon its charge [24,40]. Therefore, we aimed to block MPO uptake by pre-treatment with heparin, as previously shown [24,40,44]. MPO binding and uptake by A549 cells was 95% reduced with heparin. Furthermore, we assessed the apoptosis of cells after blocking MPO binding with heparin. The protective effect of MPO in apoptosis was abolished with heparin pre-treatment. These data suggest that MPO binding to the cell surface is a prerequisite of MPO-mediated apoptosis. 

We further investigated whether MPO can retain its enzymatic activity after intracellular uptake. First, we observed that A549 cells produce H_2_O_2_, needed for MPO activity. Using the R19S sensor, which has a high specificity for HOCl [38], we observed an increased HOCl signal in MPO-treated cells. This led us to conclude that MPO can be active after uptake into A549 cells, but this likely depends on the availability of substrate within the cell. In addition, we used the MPO inhibitor 4-ABAH [15] and assessed the impact on proliferation. Furthermore, 4-ABAH significantly reduced the effects of MPO on proliferation. These results show that the observed alterations to cellular function also depend on the enzymatic activity of MPO. Using higher concentrations of inhibitor might completely abolish the MPO effects in vitro. The involvement of MPO enzymatic activity in tumour development was also investigated in WT mice. Daily injections (i.p.) of 4-ABAH resulted in a reduction in tumour growth when compared to the DMSO-treated control mice. Moreover, 4-ABAH is an irreversible MPO inhibitor widely used in in vivo studies [15]. Further studies should include other MPO inhibitors, such as verdiperstat and AZD5904 to confirm the involvement of the enzyme activity in altering cancer cell function, both in vitro and in vivo.

## 5. Conclusions

In conclusion, our results demonstrate that MPO can directly alter cancer cell function in vitro, in a mechanism involving both binding to the cell surface and enzymatic activity. Blocking MPO from binding to cells or treatment with an MPO inhibitor reduced MPO effects on apoptosis and proliferation of lung cancer cells. Inhibiting enzymatic activity in a lung tumour engraftment model resulted in reduced tumour growth when compared to mice with active MPO. However, further studies on the mechanism of action are needed. Our data strongly suggest that MPO is an enzyme that promotes cancer cell growth and could be used as a target for lung cancer treatment.

## Figures and Tables

**Figure 1 antioxidants-12-01587-f001:**
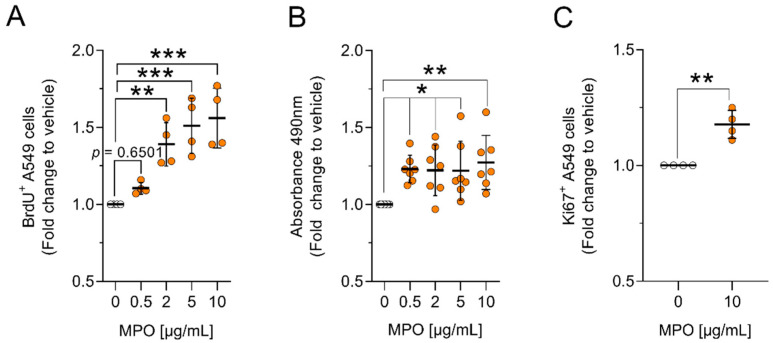
MPO increases proliferation of A549 cells in vitro. A549 cells were treated with different concentrations of MPO (from 0.5–10 µg/mL) and incubated for 48 h (**A**,**B**) or 72 h (**C**). (**A**) BrdU-assay: Data indicate mean values ± SD from four independent experiments. (**B**) MTS-assay: Data indicate mean values ± SD from seven independent experiments. (**C**) Ki67-assay: Data indicate mean values ± SD from four independent experiments. Statistical differences were assessed using one-way ANOVA with Dunnett´s post hoc test, or two-tailed t-test for comparing two groups. (0 = ddH_2_O; vehicle). * *p* < 0.05, ** *p* < 0.01, *** *p* < 0.001.

**Figure 2 antioxidants-12-01587-f002:**
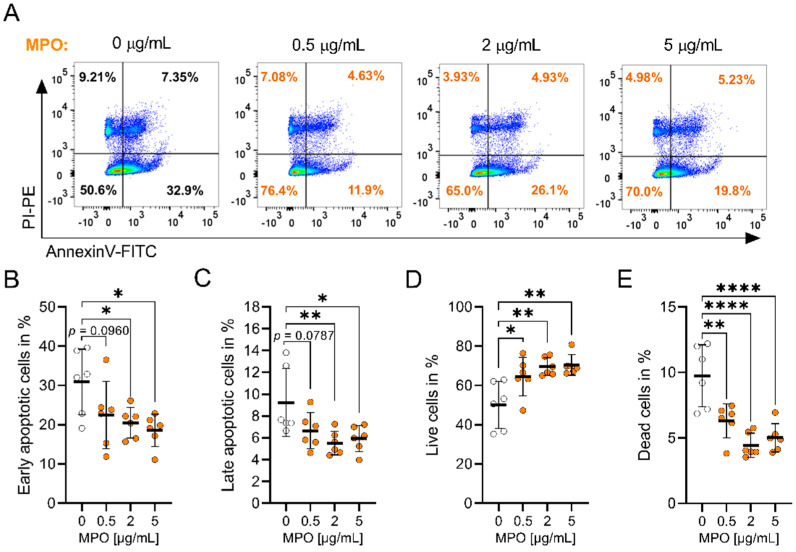
MPO reduces apoptosis in A549 cells. A549 cells were treated with MPO (in orange) for 15 h followed by Annexin V and PI staining. (**A**) Representative flow plots. Gates were set based on single colour staining of AnV and PI. Populations were distinguished as follows: (**B**) early apoptotic (AnV+/PI−), (**C**) late apoptotic (AnV+/PI+), (**D**) alive (AnV−/PI−) and (**E**) dead (AnV−/PI+) cells. Data indicate mean values ± SD from six independent experiments. Statistical differences were assessed by using one-way ANOVA with Dunnett’s post hoc test * *p* < 0.05, ** *p* < 0.01, **** *p* < 0.0001.

**Figure 3 antioxidants-12-01587-f003:**
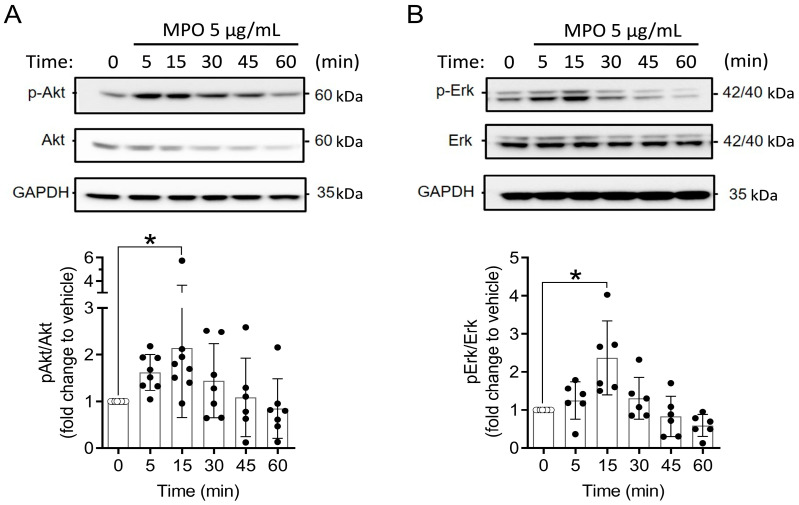
MPO increases phosphorylation of AKT and ERK in A549 cells. A549 cells were exposed to 5 µg/mL of MPO for indicated duration. Cells were lysed and analysed with Western blot analysis. (**A**) AKT and (**B**) ERK. Vehicle = ddH_2_O. Statistical differences were assessed by using one-way ANOVA with Dunnett’s post hoc test * *p* < 0.05.

**Figure 4 antioxidants-12-01587-f004:**
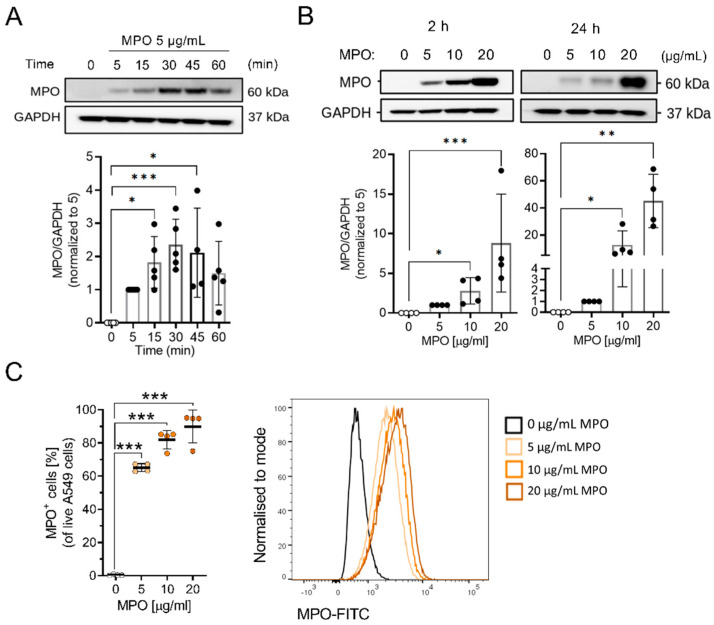
MPO binds to A549 cells. (**A**) Cell lysates from MPO-treated A549 cells for 0 (ddH_2_O = vehicle), 5, 15, 30, 45 and 60 min were analysed via Western blot analysis (N = 4–5). (**B**) Representative Western blot analysis showing MPO binding to A549 cells after 2 and 24 h of exposure (N = 4). (**C**) Intracellular flow analysis showing MPO-positive A549 cells as %. Data indicate mean values ± SD from four independent experiments. Statistical differences were assessed using one-way ANOVA with Dunnett´s post hoc test * *p* < 0.05, ** *p*< 0.01, *** *p*< 0.001.

**Figure 5 antioxidants-12-01587-f005:**
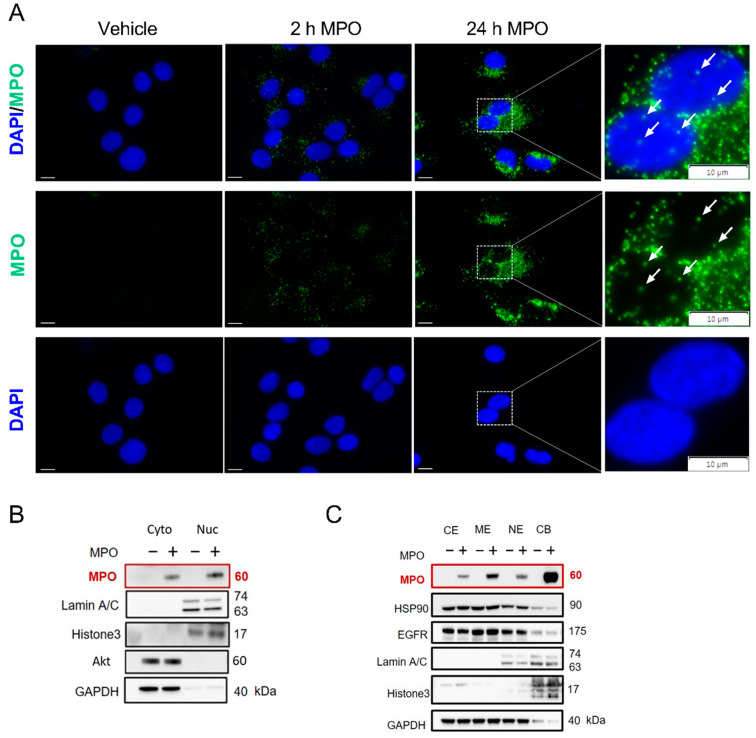
MPO internalises and translocates to the nucleus of A549 cells. (**A**) Representative immunofluorescence staining of A549 cells treated with 5 µg/mL MPO or vehicle, treated for 2 and 24 h (scale bar = 10 µm). White arrows indicate MPO signal in the nuclear region. (**B**) Representative Western blot analysis showing cytoplasmic (=Cyto, extraction volume: 500 µL) and nuclear (=Nuc, extraction volume: 50 µL) fractions of A549 cells treated with 5 µg/mL MPO for 2 h. (**C**) Subcellular fractions of A549 treated with 5 µg/mL MPO for 2 h. MPO bands are highlighted in red color. CE = cytoplasm (extraction volume: 200 µL); ME = membrane (extraction volume: 200 µL); NE = nuclear fraction (extraction volume: 100 µL); CB = chromatin fractions (extraction volume: 100 µL).

**Figure 6 antioxidants-12-01587-f006:**
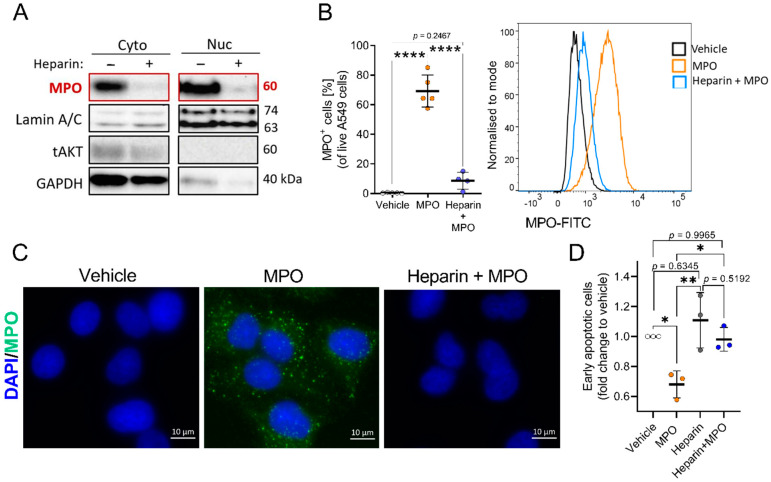
Blocking MPO binding and uptake with heparin abolishes MPO effects on apoptosis. (**A**) Representative Western blot analysis of cytoplasmic (=Cyto) and nuclear (=Nuc) fractions of A549 cells exposed to 5 µg/mL MPO for 2 h in the presence or absence of heparin (150 µg/mL). MPO bands are highlighted in red color. (**B**) Results of flow cytometry analysis of MPO-positive A549 cells after 2 h of MPO (5 µg/mL) treatment. Data indicate mean values ± SD from five independent experiments. (**C**) Representative immunofluorescence staining of MPO after 2 h of MPO (5 µg/mL) treatment ± heparin (150 µg/mL); vehicle = ddH_2_O. (**D**) Apoptosis results after 15 h of MPO exposure, showing the impact on early apoptotic (Annexin V+/PI−) cells when MPO uptake is blocked with heparin (150 µg/mL). Data indicate mean values ± SD from the independent experiments. Statistical differences were assessed using one-way ANOVA with Tukey´s post hoc test for multiple comparison * *p* < 0.05, ** *p*< 0.01, **** *p* < 0.0001.

**Figure 7 antioxidants-12-01587-f007:**
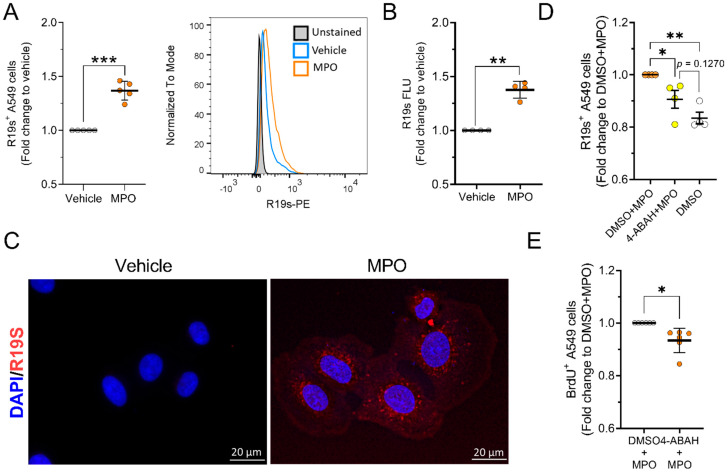
MPO preserves its enzymatic activity after internalisation. (**A**–**D**) HOCl production was measured with R19s sensor using (**A**,**D**) flow cytometry (vehicle = ddH_2_O; 0.1% DMSO; N = 4–5), (**B**) fluorescence microplate reader (N = 4) and (**C**) fluorescence microscopy (N = 1). Flow cytometry data are shown as the mean fluorescence intensity (MFI). (**E**) Proliferation of A549 cells after 48 h of treatment with 0.1% DMSO and MPO (5 µg/mL) ± MPO inhibitor 4-ABAH (10 µM) (N = 6). Statistical differences were assessed using unpaired t-test with Welch’s correction (**A**,**B**), paired *t*-test (**E**) or one-way ANOVA with Tukey´s post hoc test for multiple comparison (**D**). * *p* < 0.05, ** *p* < 0.01, *** *p* < 0.001.

**Figure 8 antioxidants-12-01587-f008:**
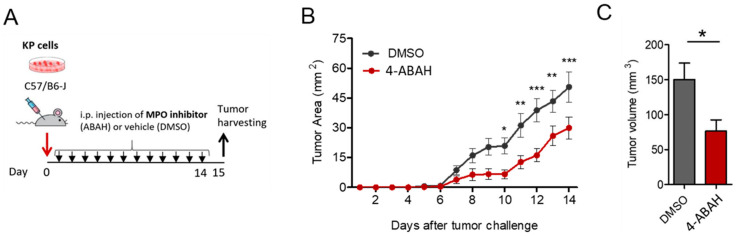
4-ABAH leads to reduction in tumour size in an engraft tumour model. (**A**) Representation of the subcutaneous (s.c.) tumour model performed in WT mice (10 mice per group). Arrows indicate treatment application and tissue harvest. (**B**) In vivo measured tumour growth. (**C**) Ex vivo measured tumour volume. All data were tested for Gaussian distribution of variables using the Shapiro–Wilk normality test. Statistical differences between the groups with normal distribution were determined using unpaired Student’s t-test with Welch´s correction, * *p* < 0.05, ** *p* < 0.01, *** *p* < 0.001.

## Data Availability

The authors declare that all the data supporting the findings of this study are available from the corresponding author on reasonable request.

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
