# Peer review of "Myeloperoxidase Alters Lung Cancer Cell Function to Benefit Their Survival"

_antioxidants, 2023, doi:10.3390/antiox12081587_

Round 1
Reviewer 1 Report
This manuscript describes interesting observations on the ability of myeloperoxidase to promote proliferation and decrease apoptosis in A459 cells. However, there aspects of the results where I consider conclusions have been too strongly drawn or where further experimental information is needed. The manuscript needs to undergo significant revision to take on board these cautions.
Major comments
1. 1. Abstract. In the light of my comments below, some statements need to toned down.
2. Figure 2. The information in the figures is highly derived from the original flow cytometry data. By normalising all the data it is difficult to appreciate the extent of the changes. More of the original data should be presented. It appears in Fig 2A that there was a lot of cell death in the cells without MPO (only 35% alive). The manuscript does not state that an inducer of apoptosis or other forms of cell death was used so why was this so high? Data on the distribution at zero time should be included, and the time given for the results in figure 2. Are the data percentages or absolute numbers? As shown, the supplement figures show very little change between 3 and 24 h, and there seems little progression from “early apoptosis” to “late apoptosis” all of which seems surprising. What is the evidence that this is apoptosis?
3. 3. Figure 4 and accompanying text. The evidence presented at this stage of the manuscript, shows that the added MPO is associated with the cells, but I am unclear of the evidence that (as stated) it is internalised. MPO attached to the surface could still be present in cell extracts. Also there seems a discrepancy between the blots, where the amount of associated MPO increases with concentration added whereas the flow data shows all concentrations gave the same result. Please explain or change the wording.
4. 4. Fig 5 shows nicely that the MPO is internalized, but the conclusion that it localises to the nucleus is equivocal. The authors need to address why microscopy shows the majority of the MPO outside the nucleus (and even where an MPO spot overlays the nucleus, can they be sure that it is internalised), yet the gels indicate most is in the nuclear extract. Could the highly positive MPO associate with negatively charged nuclear components during extraction. This possibility and the apparent discrepancy needs discussion and without more convincing evidence, it should not be concluded that the MPO localises to the nucleus.
5. 5. Figure 6. As noted in point 2, the apoptosis data needs to be shown more clearly. The 50% inhibition stated in line 495 is not apparent from the figure.
6. 6. Please give more information and be more cautious in interpreting Fig 7 A&B. First, the assay uses a kit with an unknown detection reagent, which we have to assume is detecting H2O2. Such assays are generally not well regarded. No controls are shown, such as whether it can be inhibited by catalase addition. It is stated that that the cells were incubated with probe for 15 min, so what does the 1 uM concentration refer to? What was quantified for B and what was the point to the standard H2O2. Does the signal increase over time? It is generally accepted that cells generate some H2O2, so I wonder if inclusion of this figure adds much to the manuscript.
7. 7 The fluorescence with R19-S is suggestive of MPO- generated HOCl, but the effects are modest and ideally an activity measurement on the internalised enzyme would have been more convincing. As support for the R19-S results I would include the ABAH effect in the main text. Also more information is needed on what is measured in the flow cytometry. Is it number of positive cells, or an overall fluorescence increase? How was the box in the 3rd panel of Figure S4 used? If measurements represent the number of cells in this box (an extremely small proportion), I don’t think anything can be concluded about MPO activity in the cell population.
8. 8. While Figure 8 shows that mice given ABAH show reduced tumour growth, the results are too preliminary and with insufficient control data to conclude that this effect is due to MPO. ABAH is not highly efficient at inhibiting MPO in in vivo situations. It is also not specific and reacts with other peroxidases. Important controls include showing how effectively MPO is inhibited in the mice, whether MPO is actually associated with the tumours, and whether ABAH has other effects in the mice. Ie The results are consistent with an MPO effect, but not strong enough to support a statement as in line 439.
9. Minor comments
1. Abbreviations. A number of these are used without giving the full names, especially in the Methods. Even though some are in relatively common usage, they are not familiar to some and this would improve readability.
2. It is stated that ABAH was dissolved at 40 uM in DMSO. This would mean that in experiments where 10 or 20 uM was used, the cells would be in 25 or 50% DMSO. Presumably an error?
Reviewer 2 Report
The current manuscript reports on the impact of MPO on A549 lung cancer cells and in vivo lung cancer progression. The major novel finding is that MPO can support lung cancer cell survival and progression in vitro and in vivo. In general, the manuscript is well written, and the findings interesting and clear. However, there are several major issues that require the authors careful attention.
1. It would be helpful if the levels of MPO employed can also be provided as molar concentration with an accompanying comment on how these concentrations relate to the potential in vivo concentrations of MPO in tumors.
2. In Figure 2 the impact of MPO on apoptosis are measured under basal conditions. Have the authors also studied the impact of MPO on cell viability in A549 cells in the presence of known inducers of cancer cell apoptosis?
3. The authors report that MPO activates Akt and Erk1/2 signaling. The relevance of these signaling pathways would be strengthened by studies examining the impact of Akt and Erk1/2 specific inhibitors on MPO-induced changes in proliferation and survival.
4. In Figure 6D, the effect of heparin alone on apoptosis is missing and is a critical control that needs to be included. This is because heparin can interact with numerous heparan sulfate binding proteins meaning it can potentially impact on apoptosis alone.
5. In Figure 7F, essential controls are missing, namely inclusion of parallel data for control/untreated cells and cells treated with 4-ABAH alone (in the absence of MPO). This data should be included.
6. In the mouse tumor model, have the authors measured MPO and HOCl levels in the tumour and impact of 4-ABAH on these? This is important information to validate the in vivo role of MPO in this particular tumour model and should be included.
Round 2
Reviewer 1 Report
The authors have gone some way to addressing my comments but I do still have a number of concerns.
1. Abstract. As commented below, the cell staining indicates only a very minor proportion of the MPO stain is nuclear whereas the Abstract gives the impression that it is minor. Please change. The statement “therefore …. development” (last line) is too strong and should be deleted.
2. Figure 2 description. Please elaborate on cause of apoptosis. Is it serum deprivation and is this responsible for only 50% of the cells being alive? Can the distribution of the cells with no apoptotic stimulus be given?
3. Line 356. I am puzzled by the statement that MPO reduced apoptosis at all time points, whereas it appears that only at 15 h are consistent significant differences seen. It could be considered selective to show only this time point in the main text and the others in the Supplement. At the least, the authors should acknowledge that differences were not seen at other times, and be cautious of the relevance of the one time result.
4. The growth and apoptosis results in Figs 1 and 2 show that the effect of MPO is almost maximal at 0.5 ug/ml. Yet the cell association experiments use much higher concentrations and show a concentration and time dependence. Could this mean that the effect in Figs 1 and 2 are not dependent on uptake?
5. Line 414 and Figure 5. There is clearly much more MPO outside than inside the nucleus, and it should be noted in the text that the MPO in the nucleus was only a minor fraction.
6. Differences between experiments needs to be acknowledged in interpreting results – Fig 4 shows little increase in cell – associated MPO after 15 min whereas Fig 6 shows a big increase from 2 -24 h.
7. Fig 5B&C. It would be helpful to note the differences in extract volume for the different fractions in the legend.
8. H2O2 assay. Putting this figure in the supplement does not make it more meaningful. You cannot say there is 1 uM H2O2 in the cytoplasm based on an instant response to a H2O2 standard. I still recommend removing this figure.
9. Mouse experiments. Without showing MPO is inhibited in this system, the heading should be “ABAH reduces…”. The interpretation should be “consistent with playing a role”.
10. Line 557. Relating to my comment about the timing and small fraction in the nucleus, I would remove the speculation about nuclear actions.
11. Discussion of ABAH effects. ABAH is not specific to MPO and also has been shown to be only partially effective as an inhibitor in cellular or animal studies. So please do not use “block” and “specific”.
Reviewer 2 Report
I thank the authors for their responses. They have satisfactorily addressed my concerns.
Author Response
We thank the reviewer for the constructive comments and thereby improving our manuscript.